# Arterioportal Fistulas (APFs) in Pediatric Patients: Single Center Experience with Interventional Radiological versus Conservative Management and Clinical Outcomes

**DOI:** 10.3390/jcm10122612

**Published:** 2021-06-14

**Authors:** Paolo Marra, Ludovico Dulcetta, Francesco Saverio Carbone, Roberto Agazzi, Riccardo Muglia, Pietro Andrea Bonaffini, Ezio Bonanomi, Michele Colledan, Lorenzo D’Antiga, Massimo Venturini, Sandro Sironi

**Affiliations:** 1Department of Radiology, ASST Papa Giovanni XXIII, 24127 Bergamo, Italy; l.dulcetta@campus.unimib.it (L.D.); f.carbone15@campus.unimib.it (F.S.C.); agazzi.r@tiscali.it (R.A.); rmuglia@asst-pg23.it (R.M.); pbonaffini@asst-pg23.it (P.A.B.); ssironi@asst-pg23.it (S.S.); 2School of Medicine and Post Graduate School of Diagnostic Radiology, University of Milano-Bicocca, 24127 Bergamo, Italy; 3Paediatric Intensive Care Unit, Department of Anaesthesia and Intensive Care, ASST Papa Giovanni XXIII, 24127 Bergamo, Italy; ebonanomi@asst-pg23.it; 4Liver Transplant Unit, ASST Papa Giovanni XXIII, 24127 Bergamo, Italy; mcolledan@asst-pg23.it; 5Paediatric Hepatology Gastroenterology and Transplantation, ASST Papa Giovanni XXIII, 24127 Bergamo, Italy; ldantiga@asst-pg23.it; 6Department of Diagnostic and Interventional Radiology, Circolo Hospital, Insubria University, 21100 Varese, Italy; massimo.venturini@uninsubria.it

**Keywords:** arterioportal fistula, shunt, endovascular, transhepatic, embolization

## Abstract

Arterioportal fistulas (APFs) are uncommon vascular abnormalities with a heterogeneous etiology. In pediatric orthotopic liver transplantation (OLT), APFs are frequently iatrogenic, following percutaneous liver interventions. The aim of this study was to report the 10-year experience of a tertiary referral center for pediatric OLT in the interventional radiological (IR) and conservative management of acquired APFs. A retrospective search was performed to retrieve pediatric patients (<18 years old) with a diagnosis of APF at color Doppler ultrasound (CDUS) or computed tomography angiography (CTA) from 2010 to 2020. Criteria for IR treatment were the presence of hemodynamic alterations at CDUS (resistive index <0.5; portal flow reversal) or clinical manifestations (bleeding; portal hypertension). Conservatively managed patients served as a control population. Clinical and imaging follow-up was analyzed. Twenty-three pediatric patients (median age, 4 years; interquartile range = 11 years; 15 males) with 24 APFs were retrieved. Twenty patients were OLT recipients with acquired APFs (16 iatrogenic). Twelve out of twenty-three patients were managed conservatively. The remaining 11 underwent angiography with confirmation of a shunt in 10, who underwent a total of 16 embolization procedures (14 endovascular; 2 transhepatic). Technical success was reached in 12/16 (75%) procedures. Clinical success was achieved in 8/11 (73%) patients; three clinical failures resulted in one death and two OLTs. After a median follow-up time of 42 months (range 1–107), successfully treated patients showed an improvement in hemodynamic parameters at CDUS. Conservatively managed patients showed a stable persistence of the shunts in six cases, spontaneous resolution in four, reduction in one and mild shunt increase in one. In pediatric patients undergoing liver interventions, APFs should be investigated. Although asymptomatic in most cases, IR treatment of APFs should be considered whenever hemodynamic changes are found at CDUS.

## 1. Introduction

Arterioportal fistulas (APF) are uncommon vascular abnormalities with a heterogeneous etiology [1]. They can be congenital or acquired, the latter being most frequent and related to liver neoplasms or injuries, including trauma, needle biopsy and transhepatic interventions [1,2,3,4,5]. Intra- or peri-tumoral arterioportal shunts present a high prevalence among hepatic hemangiomas [6] and hepatocellular carcinomas (HCC) [2]. APFs are also found in up to 30% of patients affected by hereditary hemorrhagic telangiectasia that causes multiple liver vascular malformations [7,8,9].

Given a different epidemiology related to age, a higher proportion of congenital APFs are found in the pediatric population, especially in infants [10,11,12,13], with a reported association with trisomy 21 [14,15]. Even though acquired APFs are less frequent in pediatric patients, their incidence in referral centers for pediatric orthotopic liver transplantation (OLT) may be increased, owing to the large number of percutaneous liver interventions performed (i.e., needle biopsies). However, due to the relative rarity and the variable etiology and clinical manifestations of acquired APFs in pediatric patients, no consensus exists on their management. To our knowledge, only several single cases and small series [16], with variable interventional radiology (IR) techniques, have been reported. Clinical guidelines with a moderate level of evidence were proposed by the Catalan Society of Digestology and the Spanish Association for the Study of the Liver [17]. They recommend embolization only for large or symptomatic shunts, with surgery reserved after a failed interventional radiology approach.

The aim of the present study is to retrospectively report the 10-year experience of a tertiary referral center for pediatric liver disease and transplantation in the IR versus conservative management of pediatric acquired APFs, with a focus on endovascular and transhepatic embolization and clinical outcomes.

## 2. Materials and Methods

### 2.1. Study Design

A retrospective search in the digital radiological report archive was performed to identify pediatric patients (<18 years old) with an imaging diagnosis of APF from January 2010 to December 2020. Patients were selected among those undergoing the post-OLT routine or urgent imaging studies, who had a diagnosis of arterioportal shunt made by color Doppler ultrasound (CDUS) or computed tomography angiography (CTA), regardless of the presence of any symptom or clinical sign. As per the institutional protocol routine, post-OLT follow-up was based on CDUS performed daily the first week after OLT, then weekly the first month and every 6–12 months thereafter. CTA was performed only when CDUS was not conclusive or in the case of urgent clinical conditions, including suspected active bleeding.

Criteria adopted to indicate IR treatment were: large APFs (>1 cm); CDUS signs of hemodynamic changes at least in the segmental vessels of the shunt: arterial resistive index (RI) < 0.5 and reversed portal flow [4]; persistence or growth of the shunt with hemodynamic changes after at least 2 weeks from the diagnosis; acute clinical manifestations or clinical signs of severe portal hypertension (hypersplenism, ascites, any grade of gastrointestinal varices). All the cases were discussed in the pediatric liver multidisciplinary team, and indications were shared with pediatric hepatologists and transplant surgeons. Patients were eligible for IR treatment depending on imaging confirmation of the above-mentioned criteria and absence of clinical or technical contraindications.

APFs were classified as types 1–3 as previously reported [1] and distinguished among congenital, acquired, iatrogenic or cryptogenic according to imaging features (position, number and size), related to clinical data (liver pathology or transplant, history of liver biopsy or other transhepatic interventions). Etiology of the shunt was gathered whenever possible. APFs were presumed to be iatrogenic when first detected after percutaneous liver procedures and not previously known. On this basis, patients without any treatment and conservatively managed were used as a control population. Written informed consent for the interventional procedures was obtained from all the patients’ parents (both mother and father) or legal guardians before treatment. The Ethical Committee of Bergamo authorized this retrospective study (Portal01; N.92/21) that was conducted in respect of the principles of the Declaration of Helsinki.

### 2.2. Interventional Radiological Management

The interventional approach (endovascular versus transhepatic) was chosen by the radiologist based on imaging findings, the elective or emergent status, APF morphology and location and particular anatomical features. The endovascular approach (Figure 1), based on arteriography and transcatheter embolization, was the first therapeutic option in acute situations. The transhepatic approach was reserved to cases of failure of the endovascular approach or when the shunt was peripheral and visible at CDUS (Figure 2).

All procedures were performed under general anesthesia: premedication with midazolam (0.5 mg/kg orally) was given in the ward; intubation was performed by dedicated pediatric anesthetists in the angiographic suite after induction with propofol 1–2 mg/kg, fentanyl (0.5–1 mcg/Kg) and rocuronium bromide (1 mg/kg); anesthesia was maintained with sevoflurane under ECG, blood pressure, pulse oximetry, temperature and capnometry monitoring. Interventional radiologists (R.A.; P.M.) with at least 5 years of experience performed the procedures with the guide of digital subtraction angiography (DSA) (Allura Xper FD20; Philips Healthcare, Best, the Netherlands).

Endovascular procedures were performed via a right transfemoral approach. A hydrophilic 4-5 Fr Cobra 2- or Simmons 1-shaped catheter (Terumo Corporation, Tokyo, Japan; Cordis Corporation, Miami Lakes, FL, USA) was placed in the celiac trunk and in the hepatic artery to perform a diagnostic angiogram; upon confirmation of the shunt, a 0.021″ or 0.027″ ID microcatheter (Carnelian, Tokai Medical Products, Kasugai-city, Aichi, Japan; Progreat, Terumo Corporation, Tokyo, Japan; Direxion, Boston Scientific, Marlborough, MA, USA) was coaxially used to navigate the intrahepatic branches on 0.014–0.018″ hydrophilic guidewires (Transend-Fathom, Boston Scientific, Marlborough, MA, USA). Superselective diagnostic angiograms were acquired to characterize the APF and to confirm feasibility of transarterial embolization. Among mechanical occlusive devices, metallic pushable or detachable 0.018″ coils (Balt Platinum coils, Balt Extrusion SAS, Montmorency, France; Interlock, Boston Scientific, Marlborough, MA, USA) and microvascular plugs (MVP, Reverse Medical Corp., Irvine, CA, USA) were used. The microcatheter tip was advanced downstream the fistula when possible, in order to perform a “sandwich” embolization of the shunt. When the APF was fed by multiple and tortuous vessels, liquid embolics such as cyanoacrylate (Glubran 2, GEM, Viareggio, Italy) or DMSO-dissolved copolymers (Onyx, Micro Therapeutics, Irvine, CA, USA; Phil, Micro Therapeutics, Irvine, CA, USA) and polyvinyl alcohol (PVA) particles (Contour Emboli, Boston Scientific, Marlborough, MA, USA) were coupled with mechanical occlusive devices.

Transhepatic procedures included both portography and direct APF embolization with liver puncture performed in the epigastrium (standard site of left lateral segment OLT). A 4 Fr coaxial introducer system (Neff Percutaneous Access Set, Cook Incorporated, Bloomington, IN, USA) was used for transhepatic portography. The segment three portal branch was accessed under ultrasound guidance with a 22G Chiba needle; the guidewire was then advanced under fluoroscopic control and the introducer pushed on the guide; a 4 Fr Cobra 2-shaped (Terumo Corporation, Tokyo, Japan) catheter was inserted through the introducer. Transhepatic direct APF embolization was performed with a 20 G Chiba needle (Chibell, Biopsybell, Mirandola, Italy): the APF nidus was targeted with the needle tip under CDUS guidance; the hemostatic matrix (Floseal, Baxter Healthcare, Zurich, Switzerland) was injected through the needle until complete APF thrombosis. The procedures were considered technically successful when the shunt was occluded or at least reduced to not hemodynamically significant, as demonstrated at final angiography for endovascular treatments or at CDUS for transhepatic approaches.

### 2.3. Conservative Management

When the criteria for IR management were not satisfied, the patients underwent imaging and clinical follow-up as per the institutional protocol.

### 2.4. Post-Procedural Imaging Follow-Up

The first post-procedural imaging control was performed with CDUS after 24 h, then after 1 week and 1 month. If no residual shunt was demonstrated, CDUS follow-up continued every 6–12 months as per the institutional protocol. In the case of APF persistence/recurrence, if the shunt was considered not hemodynamically significant and retreatment not required, routine CDUS follow-up was prescribed every 6–12 months. CTA scan was never scheduled, unless required for other clinical reasons (i.e., oncologic follow-up).

### 2.5. Clinical Evaluation and Outcome Measures

Clinical success was defined by the CDUS demonstration of shunt disappearance or reduction from hemodynamically to not hemodynamically significant: RI > 0.5 in the main arterial branch; normal flow direction in the segmental portal branch [4]. If, during imaging follow-up, the APF relapsed with hemodynamic significance (i.e., arterial RI < 0.5 and reversed portal flow in the segmental vessels of the shunt), or it determined clinical signs of portal hypertension (hypersplenism, ascites, any grade of gastrointestinal varices), clinical failure was ascribed to IR treatment. In the case of concomitant liver disease, the attribution of portal hypertension to the fistula was balanced with the overall clinical evaluation, including liver histology. Major and minor procedure-related complications were assessed, as were 30-day mortality and liver ischemia or failure. Causes of treatment failure and reintervention were analyzed in the whole study population and according to different APF features and percutaneous techniques.

### 2.6. Statistical Analysis

Continuous data are presented as the medians ± interquartile range (IQR); categorical data are given as the counts (percentage). Descriptive statistics were calculated in Microsoft Excel 2016 (Microsoft Corporation, Redmond, WA, USA).

## 3. Results

### 3.1. Study Sample

This research retrieved a total of 23 pediatric patients (median age = 4 years, IQR = 11 years; 15 males) with 24 imaging-proven diagnoses of APF. Patients’ data, including shunt features and etiology, management, follow-up and clinical outcome, are reported in Table 1.

Eleven patients underwent at least one attempt of IR treatment, while twelve patients were conservatively managed (Figure 3). Conservatively treated patients never required any kind of intervention during follow-up.

The majority of patients (20/23, 87%) were OLT recipients, and 16/20 presented iatrogenic arterioportal shunts (15 type 1; 1 type 2): etiology was most likely liver biopsy in 13 cases. The median time between biopsy and the diagnosis of the fistula was 10 days. One OLT patient developed two APFs at different times: the first was type 2, incidentally detected after OLT and probably related to surgery; the second developed as a complication of percutaneous cholangiography.

At the time of diagnosis, 4/23 patients (17%) presented clinical signs of portal hypertension; in one case, they were definitely related to the presence of the shunt. Two patients that developed an iatrogenic type 1 APF presented hemobilia and melena early after liver biopsy; both bleedings were self-limiting and did not require emergent interventions. One patient presented acute bleeding during percutaneous cholangiography and underwent emergent endovascular treatment with transcatheter embolization.

### 3.2. Imaging Diagnosis

In 14 patients, the first diagnosis of APF was made with CDUS; a total of 8/14 required further assessment with CTA. In eight patients, the shunt was first detected with CTA. DSA was directly performed in two patients: one referred from another center with a congenital APF, and one who presented acute bleeding during percutaneous cholangiography.

### 3.3. Interventional Radiological Management

Among 23 patients, 11 (48%) underwent interventional radiological treatments, either endovascular (*n* = 14) or transhepatic (*n* = 2). Two endovascular procedures ended with diagnostic DSA. Median delay from diagnosis was 9.9 days. Procedural details, reinterventions and outcomes are reported in Table 2.

Primary or secondary technical success was obtained in 12/16 procedures (75%). Technical failure in three endovascular cases was due to hepatic artery dissection, lack of exact localization of the shunt and anatomical complexity of the shunt. Technical failure in one transhepatic procedure was due to portal vein thrombosis and a lack of identification of the shunt. Five (22%) patients underwent reinterventions: in one case, the approach was converted from endovascular to transhepatic, but it was complicated and failed due to the above-mentioned portal vein thrombosis.

Clinical success was achieved in 8/11 (73%) patients with a median imaging and clinical follow-up of 42.5 months (IQR = 78.5). One patient died of multiorgan failure one month after DSA without embolization. Two patients with shunt persistence required OLT. Clinical success was also obtained in 2/11 patients that underwent DSA without embolization: in one case, APF healed spontaneously; in one case, it disappeared after iatrogenic portal vein thrombosis caused by transhepatic catheterization and a transjugular intrahepatic porto-systemic shunt.

Complications occurred in 3/18 (17%) procedures: one hepatic artery dissection that was successfully managed with balloon angioplasty; one portal vein thrombosis after transhepatic portography, which required subsequent transhepatic recanalization and a transjugular intrahepatic porto-systemic shunt; one ischemic cholangitis after transcatheter glue embolization, which required percutaneous transhepatic biliary drainage. In the first two cases, the shunt disappeared; in the last one, the APF recurred, and the patient underwent re-OLT.

The median hospital stay was variable according to the general clinical conditions of the patients. Those patients that were electively admitted to perform interventional radiological procedures were discharged after a median of 72 h after the procedure. In many cases, the embolization was carried out during longer hospitalizations required to manage medical complications of liver transplantation, and the median stay was 14 days.

Embolic devices are listed in Table 2. No differences in terms of technical or clinical success were observed with regard to the embolic device used.

### 3.4. Conservative Management

Conservative management was opted in 12/23 patients. APFs were either considered not amenable to interventional management or not clinically significant. In all but one case, the clinical outcome was good. One patient with a type 3 shunt showed persistent signs of portal hypertension.

### 3.5. Imaging Follow-Up

The imaging follow-up was only based on CDUS in 18 patients. Five patients underwent both CDUS and at least one CTA scan.

During a median imaging follow-up of 42.5 months (IQR = 78.5) after clinically successful IR management, median arterial RI improved from 0.4 to 0.6. Portal flow reversal that was present in nine cases before embolization (segmental = 5; lobar = 1; main trunk = 3) disappeared in five cases and improved from segmental and lobar to sub-segmental in two cases. In one case, portal flow reversal improved from main trunk to segmental.

The median imaging and clinical follow-up of conservatively managed patients was 42 months (IQR = 32). In these patients, median arterial RI was 0.48 at CDUS; portal flow reversal was detectable in two cases (main trunk = 1; sub-segmental = 1). In one patient with a meso-rex bypass, left intrahepatic portal flow reversal was considered physiological and remained unchanged. In five patients, the APF stably persisted over time. Four APFs disappeared spontaneously. In one patient with multiple HCC-related intrahepatic shunts, they reduced with therapy and stably persisted. In one case, a type 1 APF showed a mild increase over time at CDUS, but it was still judged not hemodynamically significant.

## 4. Discussion

Acquired APFs are usually incidentally discovered at CDUS and rarely result in acute clinical manifestations, such as bleeding, hemobilia and melena.

The physiopathology of APFs involves a direct blood flow from high-pressure hepatic arteries to lower-pressure portal veins [2,4,17], which can occur in different degrees depending on the size and location of the shunt. Guzman et al. [1] proposed the following classification of APFs: type 1 includes small peripheral intrahepatic shunts; type 2 includes large extrahepatic or centrally located intrahepatic shunts; type 3 is characterized by multiple or diffuse shunts (usually congenital or related to inherited genetic disorders). Small peripheral shunts may not produce significant hemodynamic changes in the portal system; therefore, they are usually asymptomatic and incidentally discovered, following a percutaneous liver intervention [17]. Conversely, large central fistulas and multifocal or diffuse shunts often lead to portal hypertension with secondary variceal bleeding and ascites [1,17]. As reported in the literature, acquired fistulas are most commonly type 1, while type 2 and type 3 fistulas are most likely congenital [17].

In this series, we observed a prevalence of iatrogenic type 1 fistulas. In most cases, they were caused by percutaneous liver interventions after OLT, such as routine liver biopsies. This epidemiologic trend reflects the activity of a tertiary referral center for pediatric liver transplantation. However, in a recently published retrospective analysis of more than a thousand US-guided percutaneous liver biopsies after OLT in the pediatric population, the rate of iatrogenic APF was reported to be 0.1% [18]. In the present series, the number of post-biopsy APFs was absolutely higher, closer to that observed by Falkenstein et al. [16]. This was probably due to the lower confidence with ultrasound guidance in most of the needle biopsies performed by clinicians.

Interventional radiology treatment is generally indicated for symptomatic shunts, mostly types 2 and 3, while follow-up is suggested for type 1 [17]. According to the proposed guidelines [17], type 1 fistulas tend to regress spontaneously within a few weeks from their development. Nevertheless, over a median follow-up time of about 42 months, we found that only a small number of non-hemodynamically significant fistulas regressed spontaneously. Rather, they remained unchanged over time or even exceptionally increased, therefore requiring a closer and prolonged follow-up. This behavior might be related to an underlying liver pathology that favors arterialization of the liver vasculature that feeds the shunt. In this series, about half of the fistulas showed hemodynamic significance at diagnosis and underwent early treatment. The finding of reversed blood flow in the portal pedicle afferent to the liver segment affected by the shunt was considered a worrisome feature, due to a high risk of hemodynamic impact and progression [19], regardless of the size and location of the fistula and the absence of symptoms. This is because in a small transplanted liver such as a left lateral segment OLT, even minimal alterations of portal hemodynamics can enhance portal hypertension. Indeed, the progression of small peripheral iatrogenic fistulas that become symptomatic and require treatment after months has been reported [4,20]. Some authors believe that APFs tend to progress in liver transplant recipients due to a relatively compromised arterial flow and poor compliance of transplanted grafts [4]. In this series, only one patient presented a mild increase in the shunt over a follow-up period of 36 months, but he was still asymptomatic at the time of the analysis. No conservatively treated patients became symptomatic or required interventions. The favorable clinical outcome of these patients indicates the reliability of the criteria on which the choice between interventional and conservative management was made.

The diagnosis of APF can rely on different non-invasive imaging techniques [4]. In the pediatric population, particularly in the follow-up of OLT, CDUS represents the cornerstone: it is widely available (also for the bedside approach), does not use radiation, presents a high temporal and spatial resolution for fistulas and does not require any sedation. Moreover, unlike CTA, the real-time assessment of CDUS provides information about hemodynamic changes related to arterioportal shunts, such as modification of flow direction and velocity in the portal vein up to the arterialization and modification of arterial RIs [4,21]. On the other hand, CDUS suffers from limitations such as a lack of panoramic view, limited reproducibility and, sometimes, a limited acoustic window. As per the institutional protocol, CTA is performed when vascular or extravascular complications of OLT are suspected or in the case CDUS is not conclusive in two subsequent checks within 24 h.

In our experience, the detection of reduced intrahepatic arterial RIs was a sensitive sign of APF and a reliable parameter for monitoring the efficacy of embolization during follow-up [4]. A constant finding in the case of first diagnosis or recurrence of APF was an RI of <0.5, while successfully treated patients presented normalized intrahepatic RIs [22]. Additionally, the finding of reversed flow in the portal pedicle afferent to the shunt liver segment was a sensitive marker of APF and reflected the hemodynamic impact of the shunt. These findings usually coexist, but even one can be suggestive for diagnosis. Of note, also not completely excluded APFs after one or more treatments presented a significant reduction in segmental or lobar portal flow reversal, which was considered clinically acceptable in most cases. Interestingly, most type 1 APFs of this series featured low RIs and at least segmental reversal of portal flow at diagnosis, making early treatment necessary.

Concerning the technical aspect of IR treatment, all but one APF in the current series were treated with the endovascular approach, which is the most widely used [16,17]. Only one case of a small peripheral shunt visible at CDUS was successfully treated with transhepatic US-guided injection of the hemostatic matrix. This technique has advantages in terms of its lower invasiveness and non-use of radiation and a contrast medium. However, a single treatment performed with this approach, which also seems promising, though rarely reported [23], does not allow drawing conclusions about its efficacy.

There are many aspects to consider in the IR management of APFs in pediatric patients, especially after OLT: the duration and relative invasiveness of endovascular procedures and their potential complications dictate the need for general anesthesia with close monitoring and breathing control. Patient immobility is important for safety, saving the radiation dose and the contrast medium. The variability of anatomy and the fragility of the graft require specific knowledge and skills of the interventional radiologist. All these aspects must be weighted in the decision between IR treatment versus conservative management, bearing in mind that consensus guidelines do not exist and limited reports are available in the literature. Despite dealing with the pediatric population, the techniques and materials used in this series were the standard ones for adults. However, the use of new-generation and low-profile IR devices was favored.

Transcatheter embolization was performed with different embolic materials, with coils being the most used, in accordance with the literature [24,25,26]. Microvascular plugs [27,28,29,30] were exceptionally used and, in our opinion, provided no advantages over coils. A point to consider is that to avoid shunt revascularization through collaterals, the APF feeding artery should be embolized upstream and downstream, as usually performed for aneurysms with the sandwich technique [31]. Although, theoretically, particles and liquid agents should be avoided in the case of an arteriovenous shunt due to the risk of non-target embolization, in this series, they were used, particularly in repeated interventions after failure of coil embolization. The use of gelfoam, particles and liquid embolics is described for the embolization of APFs associated with hepatocellular carcinoma [32]. Sonomura et al. [33] reported the embolization of an intrahepatic APF with an unclear site and size of the shunt by injecting glue under arterial blood flow control with a balloon catheter. In this series, particles and liquid embolics (i.e., glue, copolymers) were used without blood flow control to embolize small arterial collaterals which revascularized the shunt, provided that an obvious direct passage into the vein could not be demonstrated on angiography. Particles and liquid embolics have the advantage of penetrating small vessels distally to the microcatheter tip, allowing a more distal occlusion than coils and plugs do. They are useful in the case of peripheral APFs fed by small and tortuous vessels that cannot be catheterized to the target. From this point of view, a possible explanation of refractory APFs after coil embolization in this series may be an incomplete exclusion of the shunt, which became more and more complex in subsequent interventions due to the growth of small and tortuous collateral vessels. However, the need to resort to repeated endovascular treatments is not to be considered a failure, since a relatively high rate of APF recurrence requiring more than one treatment in up to 25% of cases has been reported [4]. Most authors agree that the successful endpoint of embolization is a reduction in the APF from hemodynamically significant to hemodynamically insignificant [4].

The transhepatic approach to the portal vein, which has been described by some authors for embolization of congenital arterioportal shunts [11] or APFs refractory to trans-arterial embolization [34], was performed in one case, but it failed. Indeed, in this series, the prevalence of OLT pediatric patients with left lateral segments provided a limited epigastric window for transhepatic portal access. This is usually performed through segment 3, which almost always coincided with the site of the fistula, limiting the space for the introducer sheath and catheter maneuver.

More complex techniques are reported for the endovascular treatment of pediatric APFs, such as the combination of stent graft and coil embolization using flow control with balloon remodeling [5], but these are probably exceptional and rarely reproduced cases and not employed in this population.

Complications occurred during or after three procedures, with a higher than the expected rate reported in the literature [26], probably due to the fragility of pediatric OLT patients. However, they were successfully managed by the interventional radiology approach, which demonstrated high tolerability.

This study has some limitations. First, it is retrospective over a wide period of time, in which the methods of diagnosis and therapy may have evolved. In addition, variable approaches and embolic devices were used without absolute selection criteria. The sample of patients was limited; in this regard, the low rate of AFP incidence has to be taken into account. Moreover, the sample itself represents a very selected case series in a tertiary referral center: this is not completely representative of the disease epidemiology in the general population but reflects the experience achieved in OLT pediatric patients.

## 5. Conclusions

Pediatric patients undergoing liver interventions should be accurately assessed for complications, including iatrogenic arterioportal shunt. Although, in many cases, they are asymptomatic and incidentally discovered, they can have a significant and potentially evolutionary hemodynamic impact, even if small. Based on this experience, we suggest the treatment of acquired fistulas which cause even minimal reversal of the portal flow, when technically feasible. A uniform technique for percutaneous embolization of APFs cannot be established because the approach is influenced by variable features. The interventional radiologist should be skilled in the diagnosis and familiar with different percutaneous approaches and embolic devices.

## Figures and Tables

**Figure 1 jcm-10-02612-f001:**
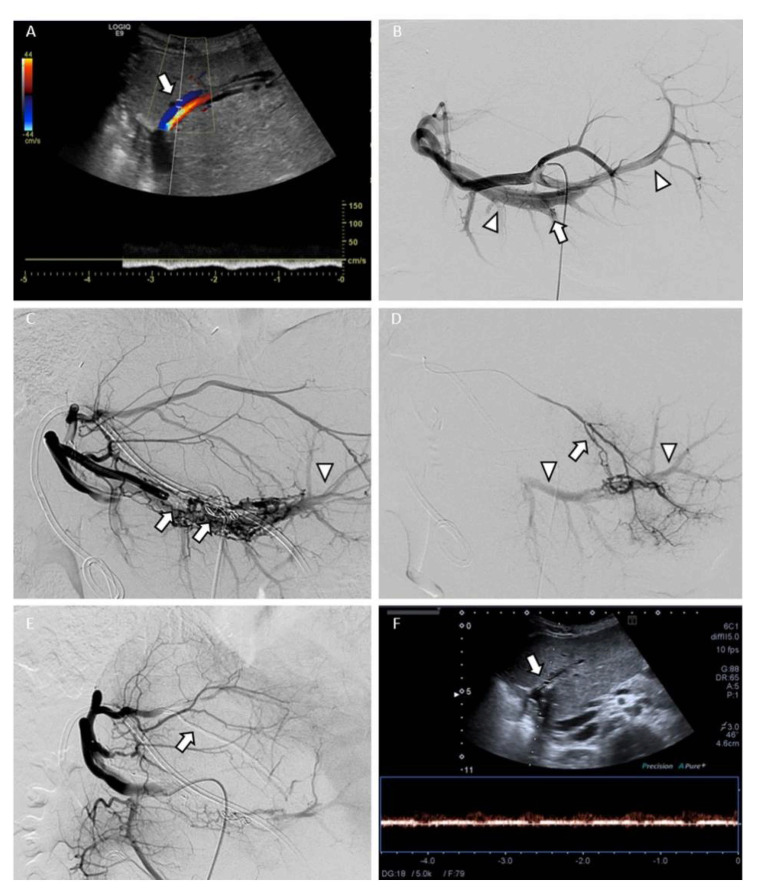
Endovascular approach: iatrogenic post-biopsy type 1 arterioportal fistula (APF) treated with transcatheter embolization in a 11-year-old child with left lateral segment orthotopic liver transplantation. (**A**) Color Doppler ultrasound (CDUS) diagnosis of hemodynamically significant APF with flow reversal in the segmental portal branch (arrow). (**B**) The celiac trunk angiogram shows the intrahepatic shunt (arrow) with abnormal opacification of the upstream and downstream segmental portal branch (arrowheads). (**C**) Complex APF revascularization through collateral vessels after embolization with microvascular plug and coil (arrows); arrowhead indicates persistent abnormal portal vein opacification. (**D**) Superselective catheterization of a collateral arterial branch (arrow); arrowheads indicate abnormal portal vein opacification. (**E**) Reduced portal vein opacification after transcatheter embolization of the collateral arterial branch with Onyx. (**F**) CDUS follow-up shows normal direction of blood flow in the segmental portal branch (arrow).

**Figure 2 jcm-10-02612-f002:**
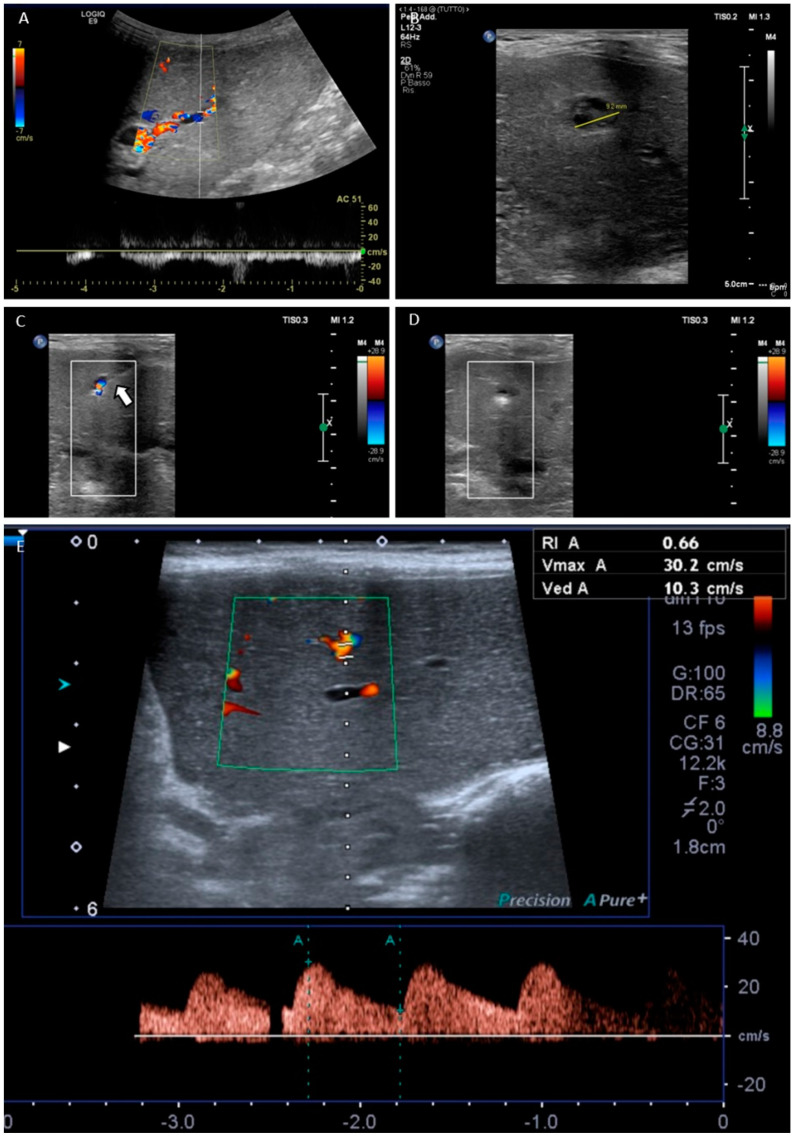
Transhepatic approach: iatrogenic post-biopsy type 1 arterioportal fistula (APF) treated with direct transhepatic hemostatic matrix injection under ultrasound guidance in a 3-year-old child with left lateral segment orthotopic liver transplantation. (**A**) Color Doppler ultrasound (CDUS) diagnosis of hemodynamically significant APF with flow reversal in the segmental portal branch. (**B**) Ultrasound visibility of a 9 mm APF. (**C**) Targeting of the APF with a 20 G Chiba needle (arrow) under CDUS guidance. (**D**) Absence of vascular signals in the APF which is thrombosed after hemostatic matrix injection. (**E**) CDUS follow-up shows disappearance of the APF with normal resistive index (0.66) in the segmental artery.

**Figure 3 jcm-10-02612-f003:**
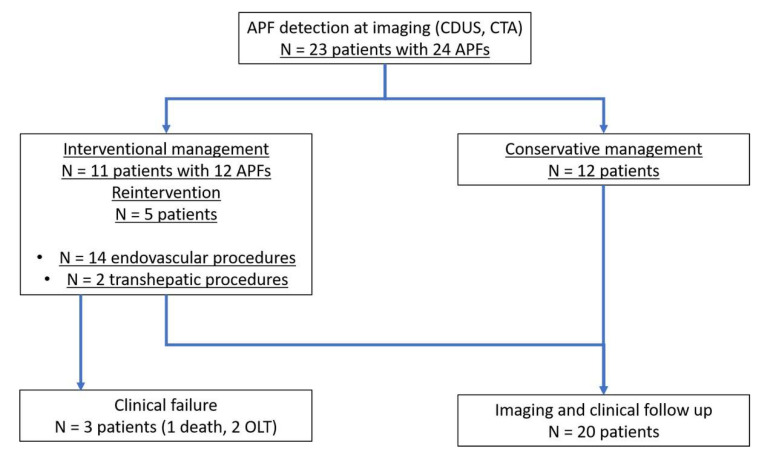
Algorithm of the study design and patient management. APF, arterioportal fistula; CDUS, color Doppler ultrasound; CTA, computed tomography angiography; OLT, orthotopic liver transplantation.

**Table 1 jcm-10-02612-t001:** Patients’ data, shunt features and general outcomes.

N.	Age	Sex	Imaging *	Shunt Size (mm)	Shunt Type	Shunt Etiology	OLT	PH	RI *	Portal Flow Reversal *	Clinical Signs	IR	FU Time (Months)	Clinical Outcome
1	11	M	CDUS	10	1	iatrogenic (biopsy)	yes	no	0.35	segmental	hemobilia	yes	9	good, APF reduction
2	2	F	CDUS	11	1	n/a	yes	no	0.35	no	no	no	36	good, APF stability
3	1	M	CTA	n/a	1	iatrogenic (biopsy)	yes	no	n/a	segmental	no	yes	92	good, APF resolution
4	13	F	CDUS/CTA	n/a	3	cryptogenic	no	yes	0.63	main trunk	no	no	78	APF persistence with PH
5	17	F	CDUS/CTA	11	1	cholangitis	yes	yes	0.4	lobar	no	yes	3	good, APF reduction
6	15	M	CDUS	n/a	1	iatrogenic (portography)	yes	n/a	0.5	no	no	no	12	good, APF stability
7	3	M	CDUS/CTA	7	1	iatrogenic (biopsy)	yes	no	0.48	segmental	no	yes	2	good, APF resolution
8	16	F	CDUS	n/a	1	n/a	yes	no	0.38	no	no	no	12	good, APF resolution
9	14	M	CDUS/CTA	9	1	iatrogenic (biopsy)	no	no	0.3	n/a (meso-rex)	no	no	36	mild APF growth
10a	4	M	CDUS/CTA	n/a	2	iatrogenic (surgery)	yes	no	0.4	segmental	no	yes	82	good, APF resolution
10b	4	M	DSA	n/a	1	iatrogenic (PTC)	yes	no	n/a	n/a	bleeding	yes	81	good, APF resolution
11	17	M	CTA	n/a	3	HCC	yes	no	0.5	no	no	no	48	good, APF reduction
12	1	F	CDUS/CTA	n/a	1	iatrogenic (biopsy)	yes	no	n/a	no	no	no	18	good, APF resolution
13	0.6	M	CTA	7	1	iatrogenic (biopsy)	yes	no	n/a	n/a	no	yes	1	death, APF stability
14	2	F	CDUS	n/a	1	iatrogenic (biopsy)	yes	no	0.46	no	no	no	75	good, APF stability
15	0.8	M	CTA	3	1	iatrogenic (biopsy)	yes	no	0.48	subsegmental	no	no	27	good, APF stability
16	0.3	M	DSA	n/a	2	congenital	no	yes	n/a	main trunk	no	no	11	APF persistence, OLT
17	3	M	CDUS/CTA	9	1	iatrogenic (biopsy)	yes	no	n/a	no	hemobilia	no	65	good, APF resolution
18	5	M	CTA	4	1	iatrogenic (biopsy)	yes	n/a	n/a	subsegmental	no	yes	87	good, APF resolution
19	4	M	CTA	5	1	iatrogenic (biopsy)	yes	n/a	n/a	main trunk	no	yes	46	APF persistence, re-OLT
20	1	M	CTA	3	1	iatrogenic (biopsy)	yes	no	n/a	no	no	no	54	good, APF resolution
21	10	M	CDUS/CTA	n/a	1	n/a	yes	yes	n/a	main trunk	no	yes	39	good, APF resolution
22	9	F	CDUS	n/a	1	iatrogenic (biopsy)	yes	n/a	0.5	no	no	no	48	good, APF stability
23	1	M	CTA	n/a	1	iatrogenic (biopsy)	yes	no	n/a	segmental	no	yes	107	good, APF resolution

* at diagnosis; CDUS, color Doppler ultrasound; CTA, computed tomography angiography; DSA, digital subtraction angiography; n/a, not available; PTC, percutaneous transhepatic cholangiography; HCC, hepatocellular carcinoma; OLT, orthotopic liver transplantation; PH, portal hypertension; RI, resistive index; IR, interventional radiology; FU, follow-up; APF, arterioportal fistula.

**Table 2 jcm-10-02612-t002:** Procedural details and outcomes of the interventional radiological management group.

N.	Emolization Technique	Days From Diagnosis	Embolic Agents	Complications	RI ^	Portal Flow Reversal ^	Primary Technical Success	Reintervention	Secondary Technical Success	Recurrence/Persistence	RI *	Portal Flow Reversal *	Clinical Success
1	endovascular, twice	15	d-coils, MVP, Onyx	none	0.35	segmental	yes	yes	yes	yes	0.4	sub-segmental	yes
3	endovascular	10	p-coils	none	0.35	segmental	yes	no	-	no	0.65	no	yes
5	endovascular, twice	10	d-coils, PVA, Phil	none	0.4	lobar	yes	yes	yes	yes	0.4	sub-segmental	yes
7	transhepatic	15	hemostatic matrix	none	0.48	segmental	yes	no	-	no	0.6	no	yes
10a	endovascular	13	p-coils	none	0.4	segmental	yes	no	-	no	0.6	no	yes
10b	endovascular	0	PVA	none	0.4	n/a	yes	no	-	no	0.6	no	yes
13	endovascular	7	none	none	n/a	n/a	n/a	no	n/a	yes	n/a	no	no, death
16	endovascular	30	none	none	n/a	main trunk	no	no	n/a	yes	n/a	main trunk	no, OLT
18	endovascular	7	p-coils	none	0.33	n/a	yes	no	-	no	0.66	no	yes
19	endovascular, three times	7	p-coils, glue	ischemic cholangitis	0.35	main trunk	yes	yes	yes	yes	0.4	segmental	no, re-OLT
21	endovascular, transhepatic	1	none	portal vein thrombosis	0.4	main trunk	no	yes	no	no	0.7	no	yes
23	endovascular, twice	4	none	hepatic artery dissection	0.45	segmental	no	yes	n/a	no	0.6	no	yes

^ pre-treatment; * at last imaging follow-up; d-coils, detachable coils; MVP, microvascular plug; p-coils, pushable coils; RI, resistive index; n/a, not available; OLT, orthotopic liver transplantation.

## Data Availability

The data presented in this study are available on request from the corresponding author. The data are not publicly available due to privacy and ethical reasons.

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
