# Peer review of "Arterioportal Fistulas (APFs) in Pediatric Patients: Single Center Experience with Interventional Radiological versus Conservative Management and Clinical Outcomes"

_jcm, 2021, doi:10.3390/jcm10122612_

Round 1

Reviewer 1 Report

Dear Authors,

Your paper describes in detail a cohort of young patients with APF and its management. The criteria to consider treatment of this APF and your results can guide clinicians to address this matter.  

Minor comments:

  • In those children with iatrogenic APF, what was the median time between the biopsy and the diagnosis of this fistula?
  • When you say cyptogenic etiology, do you mean liver cirrhosis of unknown cause?
  • Was surgical treatment considered in any of the failed cases? or would it be a possibility?

Author Response

Response to Reviewer 1 Comments

Point 1: In those children with iatrogenic APF, what was the median time between the biopsy and the diagnosis of this fistula?

Response 1: The median time between biopsy and the diagnosis of the fistula was 10 days. This sentence was added to the results section (line 214).

Point 2: When you say cyptogenic etiology, do you mean liver cirrhosis of unknown cause?

Response 2: In the manuscript the term cryptogenic was referred to a case of type 3 arterioportal fistula of unknown origin. It was detected in a child that was treated with antiviral therapy at birth due to congenital HIV infection.

Point 2: Was surgical treatment considered in any of the failed cases? or would it be a possibility?

Response 2: Surgical treatment was never performed in this case series although it could be considered a valid rescue option after a failed interventional radiological approach. However, it implies resection of the liver segment involved by the shunt, which was not be possible in these pediatric patients with two segments liver grafts.

Reviewer 2 Report

This is a clinically relevant case series describing the experience of mainly iatrogenic arterioportal fistulas in children. 

Only minor comments:

  1. What was the cause of death in patient 13? Was it related to the intervention?
  2. What were the indications of OLT in the two patients in whom clinical failure of the procedure was recorded?
  3. Post intervention management: Prophylactic antibiotics? Anticoagulation with low dose heparin for thrombosis prophylaxis? Any blood products needed? 
  4. What was the median hospital stay?

Author Response

Response to Reviewer 2 Comments

Point 1: What was the cause of death in patient 13? Was it related to the intervention?

Response 1: The patient died due to multiorgan failure which was not directly related to the intervention. Rather, death resulted from severe cardiopulmonary comorbidities.

Point 2: What were the indications of OLT in the two patients in whom clinical failure of the procedure was recorded?

Response 2: In patient 16 with a congenital type 2 arterioportal fistula the indication to OLT was severe portal hypertension. In patient 19 with a iatrogenic type 1 APF the indication to reOLT was progressive liver failure caused by chronic rejection.

Point 3: Post intervention management: Prophylactic antibiotics? Anticoagulation with low dose heparin for thrombosis prophylaxis? Any blood products needed?

Response 3: Prophylactic antibiotics were never administered as the interventional radiological procedures were considered clean.

Anticoagulation was never administered as thrombosis was the target of AFP embolization; moreover, anticoagulation could favour bleeding from site of arterial catheterization.

No bleeding required transfusions.

Point 4: What was the median hospital stay?

Response 4: The median hospital stay was variable according to general clinical conditions of the patients. Those patients that were electively admitted to perform interventional radiological procedures were discharged after a median of 72 hours after the procedure. In many cases the embolization was carried out during longer hospitalizations required to manage medical complications of liver transplant and median stay was 14 days. This paragraph was added to the result section (line 262-266).